# Second Generation of Tissue-Engineered Ligament Substitutes for Torn ACL Replacement: Adaptations for Clinical Applications

**DOI:** 10.3390/bioengineering8120206

**Published:** 2021-12-10

**Authors:** Franck Simon, Jadson Moreira-Pereira, Jean Lamontagne, Rejean Cloutier, Francine Goulet, Stéphane Chabaud

**Affiliations:** 1Centre de Recherche en Organogénèse Expérimentale—LOEX, Regenerative Medicine Division, CHU de Québec-Université Laval Research Center, Quebec, QC G1J 1Z4, Canada; franck.simon.1@ulaval.ca (F.S.); jadson.moreira-pereira.1@ulaval.ca (J.M.-P.); 2CHU Ste-Justine, 3175 Chemin de la Côte-Sainte-Catherine, Montréal, QC H3T 1C5, Canada; 3Departamento de Oncologia Clínica e Experimental, Universidade Federal de São PauloEscola Paulista de Medicina, São Paulo 04037-002, Brazil; 4Département de Chirurgie, Université Laval, Québec, QC G1K 7P4, Canada; jean.lamontagne@fmed.ulaval.ca (J.L.); rejean.cloutier@fmed.ulaval.ca (R.C.); 5Département de Réadaptation, Université Laval, Québec, QC G1K 7P4, Canada

**Keywords:** anterior cruciate ligament, collagen, goat model, endobutton, bone plug

## Abstract

The anterior cruciate ligament (ACL) of the knee joint is one of the strongest ligaments of the body and is often the target of traumatic injuries. Unfortunately, its healing potential is limited, and the surgical options for its replacement are frequently associated with clinical issues. A bioengineered ACL (bACL) was developed using a collagen matrix, seeded with autologous cells and successfully grafted and integrated into goat knee joints. We hypothesize that, in order to reduce the cost and simplify the model, an acellular bACL can be used as a substitute for a torn ACL, and bone plugs can be replaced by endobuttons to fix the bACL in situ. First, acellular bACLs were successfully grafted in the goat model with 18% recovery of ultimate tensile strength 6 months after implantation (94 N/mm^2^ vs. 520). Second, a bACL with endobuttons was produced and tested in an exvivo bovine knee model. The natural collagen scaffold of the bACL contributes to supporting host cell migration, growth and differentiation in situ post-implantation. Bone plugs were replaced by endobuttons to design a second generation of bACLs that offer more versatility as biocompatible grafts for torn ACL replacement in humans. A robust collagen bACL will allow solving therapeutic issues currently encountered by orthopedic surgeons such as donor-site morbidity, graft failure and post-traumatic osteoarthritis.

## 1. Introduction

High-pivoting sporting activities can result in a ruptured anterior cruciate ligament (ACL), with an annual incidence of 68.6 per 100,000 person years [1]. In the world, over 400,000 ACL reconstructions are required each year [2]. Affected people are mainly adolescents and young adults between 15 and 34 [3,4].Depending on the severity of the ACL injury, treatments range from nonoperative care to several surgical procedures, including autografts or allografts derived from the hamstring tendons (semitendinosus ± gracilis), the quadriceps tendon or the gold-standard bone-patellar tendon-bone (BPTB) [5,6,7,8,9]. In the USA, the cost of such surgeries is estimated at USD 18 billion [10]. Regardless of the cost of ACL reconstruction for healthcare systems, drawbacks of the current reconstruction method include the length of the recovery period [11], donor-site morbidity, frequent failure in the adolescent population [12,13,14] and a high rate of post-traumatic osteoarthritis a decade after ACL injury [15]. Better strategies to restore the stability of the knee are needed.

Tissue engineering has opened up innovative approaches to the development of tissue substitutes, ultimately reducing the need for organ donors. A reconstructed tissue is expected to become fully integrated into the host post-grafting to become permanently and efficiently integrated in situ. Thus, the scaffold of any reconstructed tissue must be strong enough to withstand physiological stresses early post-grafting and undergo remodeling to become functional. Several strategies of ACL reconstruction using tissueengineering have been described [16,17]. Among them, a successful example of a biocompatible tissue-engineered graft is the bioengineered ACL (bACL) made with a collagen scaffold. Since collagen is the natural component that supports most connective tissues, lyophilized collagen is an excellent choice to create biodegradable scaffolds [18]. After implantation into goat knee joints, the graft is colonized in situ by surrounding cells, which will reorganize its structural properties in response to the tensions and movements that it will support.

The success of bACL implantation into goats led to further studies to simplify the production process and reduce its cost, including avoiding cell isolation and cell culture, which is a costly process requiring specialized facilities, and replacing bone plugs with a more versatile attachment system. Here, we report the results of experiments that were designed to investigate the feasibility of these modifications, producing a second generation of graftable bACLs.

## 2. Materials and Methods

### 2.1. Ethics Statement

All procedures involving animals were approved by the Research Ethical Committee of Centre Hospitalier affilié (CHA) de Québec (now CHU de Québec)-Université Laval. Experimental procedures were performed in compliance with the CHA de Québec guidelines. 

### 2.2. Preparation of Acellular Graftable bACLs Anchored with Bones

A group of 12 bACLs was prepared for this experimental design. At least 3 bACLs were used for histological analyses to ensure quality control, before the surgical implantation of 3 other bACLs in the goat model. A number of 3 more bACLs were kept as a backup in case something went wrong during the surgical procedure. The last 3 bACLs were subjected to mechanical rupture tests to measure their ultimate strength values before grafting. The methods that were developed to produce and graft bACLs in goat knee joints were previously reported [18,19]. Briefly, to achieve the permanent fixation of the bACL to the bones, cylindrically shaped caprine bone plugs were prepared and pierced with a transverse hole. They were rinsed and stored in 100% ethanol for 2–3 days. A surgical thread (Maxon, size 3-0; Sherwood-Davis & Geck, St-Louis, MO, USA), absorbable within 4–6 weeks post-surgery, was passed through the holes in the two bone plugs and tied. The bones and thread were counter-rotated to provide a single twisted-thread link between the plugs. This bone/thread scaffolding was transferred to a sterile plastic tube and kept extended in a central, suspended position by passing two metal pins across the tube and through the transverse holes in the bone plugs [18,19].

For casting the bACLs, Dulbecco-Vogt modification of Eagle’s medium (DMEM) (Invitrogen, Burlington, ON, Canada) containing 10% fetal bovine serum, qualified (FBS, 12483-020, ThermoFisher, Mississauga, ON, Canada), and 1.0 mg/mL of bovine Type I collagen (isolated in our laboratory from healthy Canadian beef skin, tested for its purity by electrophoresis and solubilized in 0.1% acetic acid), were gently mixed. The mixture (total of 10 mL) was poured into 12 mL sterile plastic tubes containing the bone plugs linked by the surgical thread. The collagen polymerized in the mixture within 20 min at room temperature under a sterile culture flow hood and was maintained without any agitation. Then, the bACLs were frozen in sterile Petri dishes overnight at −70 °C and subsequentlylyophilized (Alpha 2-4 LDPlus, Martin Christ, Osterode am Harz, Germany). They were transferred back into new sterile plastic tubes and fixed again with pins before rehydration. The scaffolds’ rehydration performed in fresh DMEM produced a semirigid central core. A second coating of collagen was added as described above. Thus, within 48 h, bilayeredbACL scaffolds were obtained with a lyophilized core and an outer collagen layer. All bACLs were kept in DMEM supplemented with 10% FCS, 50 µg/mL ascorbic acid (Sigma, Burlington, ON, Canada) and antibiotics (100 U/mL penicillin (Sigma-Aldrich, Burlington, ON, Canada) and 25 µg/mL gentamicin (Schering, Pointe-Claire, QC, Canada)) until the day of the surgical implantation. Ascorbic acid promotes collagen synthesis, and adding it to collagen scaffolds could stimulate host cells to produce a de novo synthesized connective tissue matrix. Production steps are schematically illustrated in Figure 1.

### 2.3. Surgical Procedures for Implantation of Acellular bACLs Anchored with Bones into Goats

All surgical implantation procedures were performed under general anesthesia on 3 goats of 45 kg, whose native ACLs were resected at the time of implantation of the bACLs. Thus, a group of 3 acellular bACLs of the first generation was grafted in 3 goats’ knee joints. Only one leg of each goat was grafted since the other was used as a positive control.

### 2.4. Histological Analysis of bACLs before and after Implantation, Ex Vivo

Histological studies were performed on bACLs before implantation and at 6 months post-implantation. The bACL samples were fixed in an aldehyde-containing solution, embedded in paraffin, sectioned and stained by either Masson’s trichrome or hematoxylin–eosin methods to visualize the collagen matrix, the cells that had colonized the graft, including endothelial cells in blood vessels and chondrocytes. The coloration of Holmes allowed the detection of nerve endings in the grafts.

### 2.5. Mechanical Analyses

After 6 months, the bACLs will be dissected, keeping the femur–tibia system intact. All grafts and contralateral ACLs were subjected to mechanical tests (ultimate tensile strength (UTS) and stiffness). A testing machine (Instron, Corporation, Norfolk County, MA, USA) was used to measure the ultimate strength of the grafts ex vivo and post-mortem after removing all the surrounding tendons and other anatomical structures. All rupture tests (control and bACL groups) were performed at a fixed angle of 90° between the tibia and the femur. Rupturing was performed at a constant rate displacement ramp (1 cm/s) while being simultaneously recorded with a digital video camera.

### 2.6. Preparation of a Second Generation of Graftable Acellular bACLs

The first generation of bACLs was produced using a scaffold anchored with two bone plugs [18,19,20,21,22,23,24]. However, the second generation of bACLsdid not contain bone anchors. The new bACLswere fixed in situ using the Endobutton CL BTB Fixation Device, or cortical button (Smith & Nephew, Andover, MA, USA). It provides strong, dependable ACL fixation and eliminates the need for knot tying, providing a strong and stiff repair.

A braided thread scaffold (sterile braided Vicryl Polyglactin 910 No. 3, CCS-1, coated, Ethicon Inc./Johnson & Johnson, Markham, ON, Canada), absorbable within 4–6 weeks in situ post-surgery, was passed through an endobutton. The thread itself is braided, but to further reinforce the bACL scaffold, three threads were braided together and used as a central core. The thread scaffold was transferred to a long sterile plastic tube and kept extended in a central, suspended position. The endobutton’s weight pulled the scaffold towards the bottom of the tube and kept it in place. Once the thread scaffold was ready, native Type I collagen was poured around it, filling the whole casting tube. 

To cast the bACLs, DMEM containing FBS and 1.0 mg/mL of bovine Type I collagen (solubilized in acetic acid 0.1%) was readily but gently mixed. The mixture (total of 25 mL) was poured into a 30 mL sterile plastic tube containing the surgical braided threads. The collagen polymerized within 20 min at room temperature under a sterile culture flow hood and was maintained without any agitation. Following polymerization, the collagen becomes a gel. To prevent the gel from drying, it must be covered with about 2 to 3 mL of DMEM supplemented with 10% FBS, 50 µg/mL ascorbic acid and antibiotics. A cap was added to cover each tube, and the scaffolds were placed in an incubator in an atmosphere containing 8% CO_2_.

All the bACLs were frozen in sterile Petri dishes overnight at −70 °C and subsequently lyophilized. Then, they were transferred into new sterile plastic tubes. The following rehydration in fresh DMEM to produce a semirigid central core, a second coating of collagen, could be performed as described above. BilayeredbACLs contained a central lyophilized and rehydrated core. The acellular bACLs were used for surgical testing of the endobutton’s fixation technique in vitro.

A group of 3 bACLs wascellularized by seeding a suspension of dermal fibroblasts (DFs, at a concentration of 2.5 × 10^5^ cells per mL) in the second collagen layer. This group was produced to test the biocompatibility of the braided threads’ core and confirm that the cells could progressively adhere to the braided threads and contract the outer collagen layer in vitro over 24 h and thereafter. The bACLs containing DFs were sent for histological analyses to see if the collagen fibers and the living cells were aligned in the same direction with the braided thread core. 

### 2.7. Surgical Testing

The technical modifications of the bACL implantation procedure were assessed in vitro on fresh bones of bovine knee joints. The approach of implantation using endobutton fixations has become widely used in orthopedic surgery. However, our bACL is a unique ACL substitute, produced entirely in vitro. It was important to assess the feasibility of the bACLs implantation using endobutton devices. The use of a protective envelope during the passage of the graft in the bone tunnels had to be tested as well. Thus, 3 bACLs were implanted into fresh bovine knee bones in vitro to assess the fixation procedures. This was qualitative testing realized by two independent, well-qualified orthopedic surgeons. The objectives were to assess the resistance of bACLs to manipulation in a simulated surgical condition on a joint similar to a human joint. Handling resistance inside bone tunnels and tensile strength when fitting and installing the bACL. The macroscopic aspect of the graft was examined by the surgeons.

### 2.8. Mechanical Characterization of the Braided Thread Scaffold

Braided Vicryl threads (*n* = 3) were subjected to uniaxial tensile testing on an Instron ElectroPuls E1000 mechanical tester (Instron Norwood, Norfolk County, MA, USA). Rupture assays were performed on braided Vicryl scaffolds. Both extremities of the specimen were stretched at a constant rate of 0.2 mm/s until the tissue ruptured. Data were analyzed using Minitab v17 (Minitab, State College, PA, USA) to provide ultimate tensile strength and stiffness.

### 2.9. Histological Analyses of the Scaffold

It was interesting to analyze the central braided thread core of the bACL after lyophilization and rehydration to assess the quality of the collagen matrix surrounding the braid. This work was performed by a private company, equipped to cut hard tissues such as surgical thread, into ultrathin slices of 10–15 µm-thick.

## 3. Results

### 3.1. Grafting of the First Generation of Acellular bACLs

The first generation of bACLs was designed according to the bone-patellar tendon-bone (BPTB) graft, as it consists of tendon and bony attachments. Bones were attached to the bACL scaffolds. The grafting of such tissue-engineered grafts was performed the same way it is performed in humans. Figure 2A shows one of the first-generation bACLs of that was grafted successfully in the goat model for 6 months. The bACLs were produced without cells, using native bovine Type I collagen as a matrix. The bACLs were cultured under minimal tension, and the collagen fibers became aligned in a direction parallel to the tension applied (Figure 2B). Endothelial cells colonized the bACLs in situ, as shown by the blood vessels (Figure 2C; note the red or pink endothelial cells). The graft also contained nerve endings (Figure 2D; brown structures and fibers). Additionally, chondrocytes were observed at the interface between the ligament scaffold and each bone anchor (Figure 2E). 

After 6 months, the acellular graft was populated with cells, vascularized, innervated and reinforced in situ (Figure 3A), showing a macroscopic aspect that is very close to a native ACL (Figure 3B). The bACL grafted in situ for 6 months reached an average of 18% (94 ± 14 N) of the native contralateral ACLs (520 ± 59 N) (Figure 4, Table 1). 

### 3.2. Endobutton Fixations of bACLs

When anchored with bones, the diameter and the length of the bone plugs had to be relatively precise to respect the anatomic features of the knee joint. The length of the bACL also had to be measured. However, the length of the second generation of the bACL can be variable, as it is pulled and then attached conveniently with the endobutton’s fixation. This makes the use of such bioengineered grafts adaptable to several sizes of knees. From its femoral attachment, the ACL has a length that ranges from 22 to 41 mm (mean, 32 mm) and its width from 7 to 12 mm [25]. When a bACLis cast (Figure 5A), the braided thread is placed in a sterile tube under the flow hood, and the weight of the endobutton pulls it at the bottom, while the solution of collagen is poured in the tube. After about 20 min, the collagen had polymerized (Figure 5B and Figure 6A). It could be readily frozen and lyophilized until use (Figure 6B). Once rehydrated (Figure 6C), an acellular bACL could be placed in the cylindrical plastic tube and stored at 4 °C. Some histological analyses of the scaffold revealed that the collagen adheres to the central absorbable braided thread (Figure 6D,E). Collagen fibers are aligned in the direction of the tension applied to the tissue (Figure 6D,E). 

### 3.3. Strength of the Braided Thread

A rupture assay was performed on a group of braided threads to characterize their properties. The stiffness and UTS are important parameters to know (Table 1). The mean ultimate load in specimens aged 22–35 years was 2160 (±157) N. For the specimens aged 22–35 years, the stiffness was found to be 242 (±28) N/mm [25].

The complex geometrical configuration and different-length fiber bundles of the ACL have hindered efforts to calculate stress and strain. Butler et al. divided the human ACL ligament into portions and tested the individual units for average modulus and ultimate tensile strength. The average ultimate tensile strength measured 278 and 35 N/mm^2^, respectively [26]. The ligaments reached their ultimate stress at −15% strain.

The procedure of implantation was tested in vitro by two orthopedic surgeons (Figure 7A–H). The technique that was used is the same that the orthopedic surgeons chose, using the doubled semitendinosus/gracilis autograft. The bACLs have to measure between 18 and 25 cm to allow the possibility to fold them twice and adjust their length to the specific features of the knee joint. The bACL was protected with a semipermeable membrane i.e., dialysis tubing.

## 4. Discussion

Several parameters must be considered to produce a graftable ligament substitute. The biocompatibility of the bACL will determine its level of integration in the bone knee joint of the host. The ultrastructural features of its scaffold play a critical role in cell colonization, before and/or post-implantation. The biomechanical properties of the bACL must resist the physiological stress that will be applied to its structure during knee movements in situ. Torn ACL replacement is a challenge since the ACL plays an important role as a major knee joint stabilizer, being subjected to a lot of stress, notably because of its anatomic location [27,28]. 

Present tissue-engineered ACLs are mainly produced using biomaterials such as polycaprolactone [29], poly (glycolic acid) (PGA) [30], poly(lactic-co-glycolic acid) (PLGA) [31], poly(L-lactic acid) (PLLA) [31,32], or silk [33,34]. All these biomaterials are biodegradable but at various rates not necessarily compatible with adequate regeneration of the ligament. Additionally, with the exception of silk, their degradation products include lactic acid, which can have a negative role, as demonstrated by its effect on cancer progression [35]. However, these materials have tunable mechanical properties and can be chemically modified to recreate a more physiological environment than the raw materials (e.g., [31]).

A collagen scaffold, anchored with two bone plugs and seeded with autologous ACL fibroblasts, was developed in vitro and successfully grafted in several groups of caprine knee joints to replace a torn ACL invivo [18,20,21,22,23,24]. An absorbable surgical thread was added to the collagen matrix to contribute to its initial structural reinforcement and facilitate its manipulation during implantation [18]. In the first generation of bACL models, bone plugs were used to fix both ends of the graft, adding screws in the bone tunnels to fix the plugs in situ. Collagen remodeling must occur in the bACL scaffold to become a functional ACL [18]. The collagen scaffold promotes cell migration, growth and differentiation, and the first bACLs have shown excellent results in the goat model [18,19]. One month post grafting, blood vessels are visible. At 3 months, chondrocytes can be observed and Sharpey’s fibers connect the graft to the bone anchors. Some nerve endings are observed on histological sections of the graft stained with the method of Holmes. Slowly, the bACL is permanently linked to the knee joint. After a year, it is hard to differentiate the graft from the contralateral native ACL in the goat knee joints [18]. The bACL integrated in situ reached an average of 36% (+5%) of native ACL strength after only 13 months, without any specific training program applied on the goats post-surgery. At this step of the bACL development, two issues remain to simplify the production process and reduce cost and regulatory obstacles: cells and bones. This is the subject of the present study (Table 2).

Most of the first-generation bACLs that were grafted in the goat model contained living autologous ligament or dermal fibroblasts [18,19]. The advantage of adding living fibroblasts in the graft before implantation was the early initiation of caprine collagen synthesis and remodeling, slowly replacing the bovine collagen fibers. Thus, autologous cell seeding is always suitable to produce autologous ACL substitutes if needed. All living fibroblasts secrete collagen, and they also contract collagen fibers, gathering fibrils, to produce a denser matrix. Living cells contribute to gathering matrix fibers around the central core made of absorbable braided threads, before freezing and lyophilization of the bACL that kill the cells. Once rehydrated, the matrix regains about 60–75% of its initial volume. At this step, a choice can be made by adding or not autologous fibroblasts in the outer layer of the reconstructed tissues. The autologous fibroblasts migrate into the bACL matrix, proliferate and initiate collagen remodeling thereafter. Thus, this option is not essential, but it remains advantageous in some specific clinical contexts. However, to simplify bACL production, the integration potential of acellular grafts was assessed on the caprine model for 6 months. The results confirmed that such an approach was feasible to replace a torn ACL. The acellular grafts became cellularized and reinforced within a month in situ post-implantation. However, acellular grafts can also be integrated into a knee joint. The fibroblasts of the host slowly populate the graft in vivo, secreting matrix components and growth factors that may attract other cell types, such as endothelial cells. 

The use of bone anchors works very well, but it also creates a need for bone samples that may eventually limit the feasibility of the tissue-engineering approach in addition to the risks of disease transmission associated with any allograft tissue. It also considerably increases the cost of bACL production (in time and native bone materials processing). The bACL would gain to be adapted to be shipped and grafted in any surgical room, anywhere in the world. The initial bACL collagen scaffold remains fragile to shocks during implantation. Passing the graft through the tunnels performed in the tibial plateau and the femoral condyle causes wear. Thus, the bACL must be protected during its technical insertion in situ. The second generation of the bACL will be held in the knee joint by the endobutton fixed on the femoral condyle and a tibial interference screw in the tibial bone. The endobutton fixation device has shown good results in the dog knee joint [36,37]. This approach will have to be assessed in vivo on the goat model, but the scaffold will be made with the same collagen matrix around a central core of absorbable braided thread, so the results are expected to be excellent too. The use of endobutton fixations to replace the bone–ligament–bone technique of graft implantation simplifies the whole concept of tissue-engineered ACL substitutes as an option for torn ACL replacement. The surgical tests performed in vitro had to be performed before planning an experiment conducted on animal knee joints in vivo. The tests performed in vitro, by two orthopedic surgeons recognized as experts in ACL reconstruction, confirmed that the bACL of the second generation could be grafted and fixed without bone anchors and would have the strength to resist the endobutton’s fixation. This modification of the procedure would avoid the need for a bone source. Moreover, despite the neoformation of ultrastructured cartilage, including Sharpey’s fibers anchored to the knee bones of the host post-grafting, the interface between the bone plugs and the ligament substance of the first generation of the bACL was its weakest link. 

From a technical point of view, the insertion of the bACL in the bone tunnels performed in the femur and tibia during implantation remains a delicate step, since the implant is rubbed against the sharp edges of the osseous tunnels during sliding. The use of sterile dialysis tubing to protect the implant during its insertion in the bone tunnels solved this technical issue. It could also serve at wrapping bACLs for shipping them to various laboratories for eventual preclinical trials. 

An interesting improvement in our production process can be the replacement of bovine TypeI collagen by its recombinant human counterpart, which is now commercially available. Our protocol for collagen extraction and purification involves many steps of solubilization and precipitation of the collagen protein and a lyophilization step. These steps collectively ensure the complete removal of noncollagenic elements. However, the use of recombinant human TypeI collagen can solve many regulatory issues, but with an increased cost.

The choice of the braided structure, i.e., braided Vicryl^®^ (polyglactin 910) surgical thread, relied on its sufficient mechanical properties to sustain the pressure exerted on its structure during the first week post-grating (Table 1) and on its biocompatibility and potential positive effect in tissue regeneration [38].

For now, the bACL fills the requirements of integration ability, low cost and simple production process. Nevertheless, in the future, depending on the clinical context, several optimization steps could be performed to increase the potential of the bACL. Interestingly, collagen can be mixed with other matrix components and/or growth factors to stimulate cell migration, growth and differentiation [39,40]. Penkova [41] reported that in the presence of glycerol, the collagen molecule was stabilized, not only by heating but also by the action of urea. This report is in agreement with an interesting observation made in our laboratory that could also contribute to improving the rigidity and ultimate strength of a tissue-engineered tissue. When a bACL was dipped into a fresh solution of glycerol (from a concentration of 0.1% in DME or above), the tissue became more resistant to rupture, and living fibroblasts could migrate into the collagen scaffold of the bACL, then contract the matrix, as observed in untreated bACLs in vitro [42]. Work is required to evaluate with precision the percentage of the gain in strength induced by the glycerol on collagen fibers. Electron microscopic analyses showed that glycerol formed a coating on the collagen fibers, without altering their three-dimensional organization (data not shown). Additionally, the interesting work of Benson et al. [43] demonstrated that a suture tape augmentation acted as a stabilizer during the early stages of the graft incorporation.

Interestingly, major efforts have recently been made to effectively reduce the use of laboratory animals and comply with the 3Rs policy [44]. Among these efforts, the computer modeling of various parameters makes it possible to obtain a great deal of information related to biomaterials and their behavior in physiological environments [45,46,47,48]. In the future, an in-depth analysis of our bACLs with bioinformatics tools could make it possible to reduce the number of animals required and therefore both the ethical problems related to them and the costs of the studies to be carried out. However, regulatory agencies still request that tests be carried out on animal models because certain parameters remain difficult to assess in silico.

In this study, three goats were successfully transplanted to demonstrate that acellular bACLis an acceptable treatment option. Tests were also carried out ex vivo on bovine joints with second-generation acellular bACLs to evaluate the effect of replacing bone plugs with endobuttons. The number of animals used is low, and this study should be considered as proof of principle that paves the way for further characterization. The idea of going step by step is to reduce the number of animals used in our study by eliminating potentially failing conditions. As previously stated, we hope that using a computer model will keep the number of animals that will need to be implanted to a minimum.

## 5. Conclusions

In conclusion, our technology developed to produce graftable ACL substitutes seems very promising. These data are good indicators of the potential that can be explored to produce and graft bioengineered ACL substitutes permanently and shortly. This is a powerful tool for torn ACL replacement.

## Figures and Tables

**Figure 1 bioengineering-08-00206-f001:**
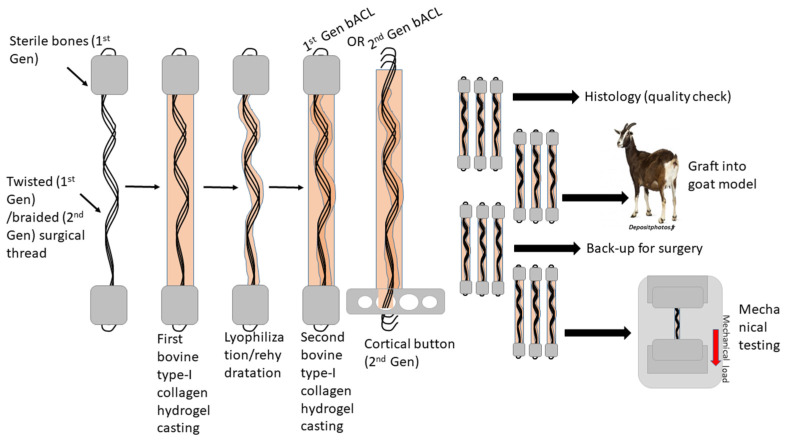
Schematic illustration of the production process of bACLs.

**Figure 2 bioengineering-08-00206-f002:**
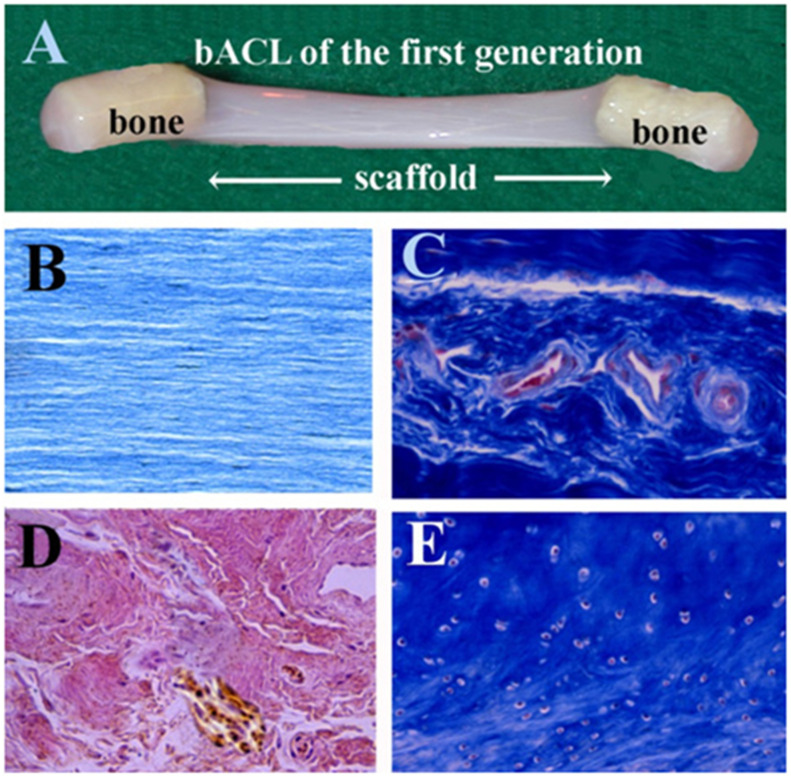
Macroscopic view and histological feature of the first-generation bACL. (**A**) A bACL of the first generation. (**B**)The bACLs were produced without cells, using native bovine Type I collagen as the matrix. (**C**) After implantation, the bACL became vascularized (red or pink endothelial cells) and (**D**) also contained nerve endings (brown structures and fibers). Additionally, (**E**) chondrocytes were observed at the interface between the ligament scaffold and each bone anchor. (×40).

**Figure 3 bioengineering-08-00206-f003:**
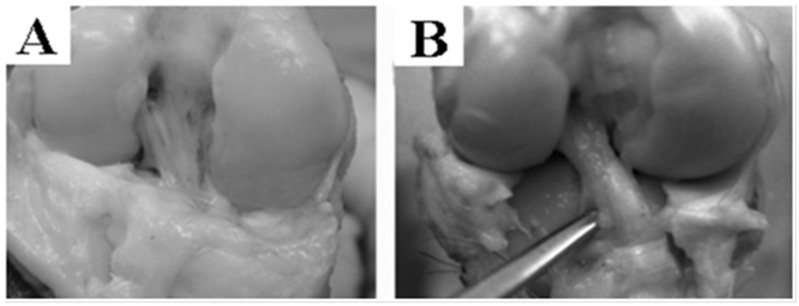
Macroscopic view of bACLs after implantation. A bACL of the first generation was grafted successfully in the goat model for 6 months. The bACLs were produced without cells, using native bovine Type I collagen as the matrix. After 6 months, the graft was remodeled, vascularized, innervated and reinforced in situ (**A**), showing a macroscopic aspect that is very close to a native ACL (**B**).

**Figure 4 bioengineering-08-00206-f004:**
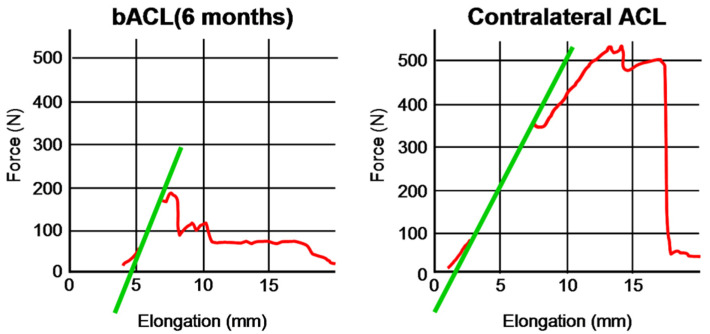
Elongation curves of bACLs and contralateral ACLs tested after 6 months of implantation into goats. Representative curves for bACLs (**left**) and ACLs (**right**).

**Figure 5 bioengineering-08-00206-f005:**
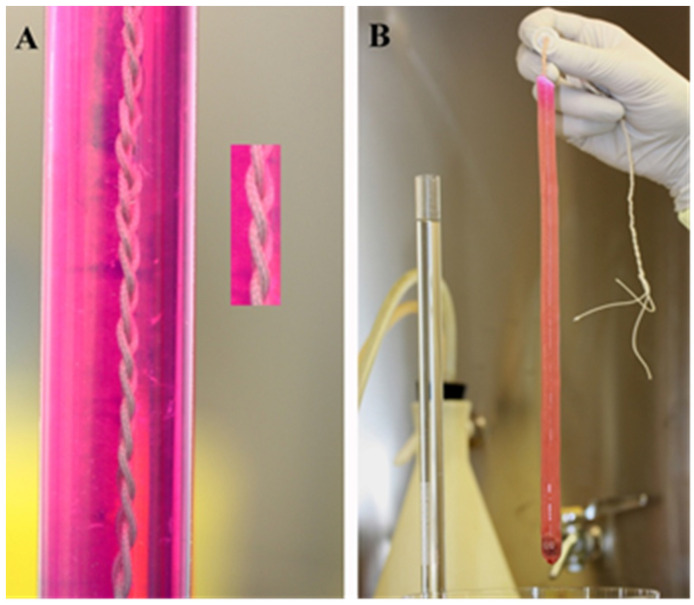
Macroscopic view of the casting step of bACL. (**A**) A picture showing the casting step of a bACL of the second generation, using absorbable surgical thread braided and placed in a cylinder of 30 cm high and 10 cm diameter, filled with a solution of 2 mg/mL Type I bovine collagen that polymerized around the thread (**B**). Note that the collagen matrix has become a gel within 20 min at room temperature that can be manipulated outside its casting tube.

**Figure 6 bioengineering-08-00206-f006:**
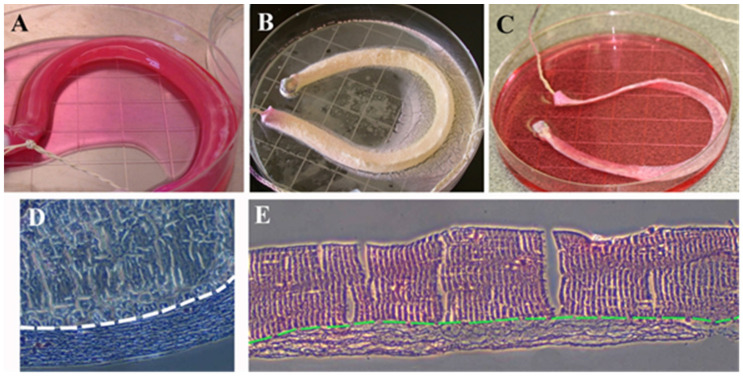
Macroscopic view of lyophilization steps of the bACL and its histological features. (**A**) A bACL of the second generation made of native Type I collagen matrix polymerized around the braided thread scaffold before (**B**), its lyophilization at −80 °C and (**C**) its rehydration at 4 °C. Histological sections of the thread surrounded by Type I collagen were stained using the Trichrome de Masson’s (**D**) and the hematoxylin–eosin techniques (**E**). In (**D**,**E**), the structural aspect of the thread is shown above the white or the green dashed line, while the collagen seeded with ACL fibroblasts can be observed below the line (×40).

**Figure 7 bioengineering-08-00206-f007:**
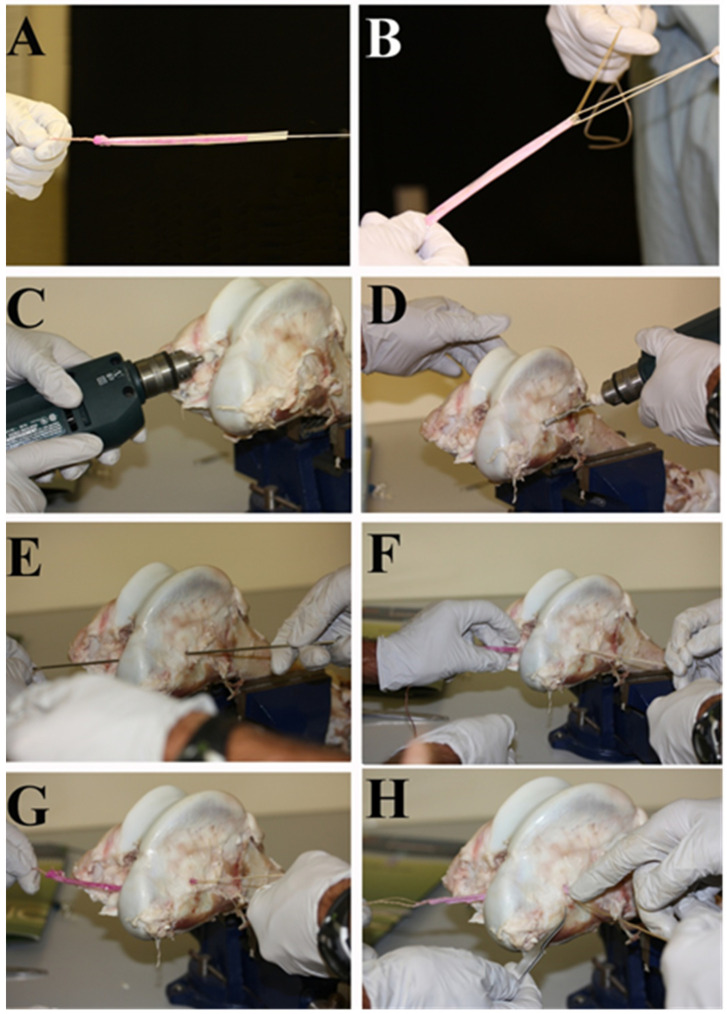
Second-generation bACL surgical testing. (**A**) A bACL of the second generation held by braided surgical threads integrated into its collagen scaffold, (**B**) the orthopedic surgeons apply tensile strength on the bACL to make sure it canbe easily manipulated, (**C**) the first tunnel is drilled into the tibia, (**D**) the second tunnel is drilled in the femoral lateral condyle, (**E**) the surgeon passes a metal rod through the tunnels to attach the bACL to it and pull it, (**F**) the bACL has gone through the tibial tunnel and then through the femoral tunnel (**G**), tension is applied on the implant (**H**) and it is ready to fix with an endobutton.

**Table 1 bioengineering-08-00206-t001:** Mechanical features of acellular bACLs grafted for 6 months in the goat model, Vicryl braided thread used in the bACL scaffold in vitro and native ACLs used as controls.

Specimen Tested	Ultimate Tensile Strength (N/mm^2^)	Stiffness (N/mm)
ACL (a)	520 ± 68	64 ± 11
bACL (a)	94 ± 14	102 ± 15
Vycril braided thread (b)	340 ± 28	10.2 ± 0.3

(a) Rupture test; (b) uniaxial tensile test.

**Table 2 bioengineering-08-00206-t002:** Summaryof differences between 1st and 2nd generation acellular bACLs.

	1st Generation of Acellular bACLs	2nd Generation of Acellular bACLs
Matrix	Bovine Type I collagen hydrogel (can easily be replaced by commercially available recombinant human Type I hydrogels)	idem
Anchorage	Sterile bone plugs	Endobutton (cortical button)
	(limited availability and potential regulatory issues)	
Protection of the graft during implantation	None	Dialysis membrane (efficiency tested by orthopedic surgeons)
Advantages	Surgical procedure similar to standard BPTB procedure	All items used are commercially available and approved by the FDA

## Data Availability

Data are available upon reasonable request from the corresponding authors.

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
