# Peer review of "Second Generation of Tissue-Engineered Ligament Substitutes for Torn ACL Replacement: Adaptations for Clinical Applications"

_bioengineering, 2021, doi:10.3390/bioengineering8120206_

Round 1
Reviewer 1 Report
In this manuscript, the acellular anterior cruciate ligament (ACL) grafts were engineered as economic and simplified models and then studied to adapt for clinical applications. The presented work is useful and could be employed by further investigations to provide perfect scaffolds for this application. Obviously, the authors put much efforts writing this manuscript which is systematically and comprehensively elaborated. However, there are still some issues to be addressed. The followings are my comments:
- It would be valuable to show the preparation process of scaffolds provided by the present study in one figure, maybe as a schematic illustration or others, to clarify this section.
- The "Grafting of the first generation of acellular bACLs" (P. 4, line 182) should be displayed in a bold format.
- It is suggested to apply the morphological characterization of different scaffolds just after preparation to show and subsequently evaluate the physical properties such as pores, different layers, etc. To this aim, scanning electron microscopy (SEM) observation is a good technique.
- One type of scaffold used in this study is based on the braided structure but there is no supporting discussion that why this specific structure was selected in this work.
- It has been mentioned that host cells synthesized the matrix components (P. 10, line 315). Are there any results that supported this claim?
- The authors did not discuss the results obtained from mechanical testing.
- The desirable effect of the scaffold on the host cells was pointed out; "The natural collagen scaffold of the bACL contributes to support host cell migration, growth, and differentiation in situ post-implantation. (abstract section)" But there are no related results and discussion that approved this observation, especially in the terms of cell differentiation.
- It is suggested to add a comprehensive review to summarize and discuss two prepared scaffolds, first and second generation of acellular graftable bACLs, their differences, advantages and disadvantages, etc.
- Both grafts investigated in this study are acellular grafts. It would be better to mention the decellularization process and also its efficiency by DNA quantification or other methods (if possible).
- Although the researchers applied the control group, there is no comparison between the results obtained from the grafts and controls.
- The "Conclusion" part and an appropriate description should be added.
Reviewer 2 Report
General comment
The paper aims to design and validate tissue-engineered ligament substitutes for torn ACL replacement.
The rupture of the anterior cruciate ligament is quite common (about 400,000 per year), suggesting the relevance of the investigation.
The common treatment of CL rupture consists in autograft or allograft derived elements from tendons. Such a procedure demonstrated several limitations and drawbacks.
The here proposed approach account for the methods of tissue engineering, such as the implantation of a collagenous graft, which is subsequently colonized in situ cells, which will remodel the biomechanical properties of the novel structure depending on tensions and movements.
The abstract must be modified including more quantitative results of the study.
The methods for the preparation and implantation of the bACL samples are well reported. The characteristics of the bACL are evaluated 6 months after the implantation. However, the mechanical behavior has not been adequately investigated. Mechanical properties of ACL are evaluated considering stiffness and ultimate strength only, and no data were reported on experimental setup (strain rate…). On the other side, ligaments are complex structures with anisotropic and non-linear elastic response. The authors are encouraged to report stress-strain curves and data about mechanical properties along different directions.
A major drawback of the here proposed activities pertains to the use of animal models, and the subsequent ethical costs. “In silico” techniques may provide reliable support to replace or to minimize experimentations on animal models. The Authors are encouraged to discuss about this topic.
Forestiero, A., Carniel, E.L., Natali, A.N., Biomechanical behaviour of ankle ligaments: Constitutive formulation and numerical modelling (2014) Computer Methods in Biomechanics and Biomedical Engineering, 17 (4), pp. 395-404.
Orozco, G.A., Tanska, P., Mononen, M.E. et al. The effect of constitutive representations and structural constituents of ligaments on knee joint mechanics. Sci Rep 8, 2323 (2018). https://doi.org/10.1038/s41598-018-20739-w
Benos Lefteris, Stanev Dimitar, Spyrou Leonidas, Moustakas Konstantinos, Tsaopoulos Dimitrios E. A Review on Finite Element Modeling and Simulation of the Anterior Cruciate Ligament Reconstruction . Frontiers in Bioengineering and Biotechnology, 2020. https://www.frontiersin.org/article/10.3389/fbioe.2020.00967
Forestiero, A., Carniel, E.L., Fontanella, C.G., Natali, A.N., Numerical model for healthy and injured ankle ligaments (2017) Australasian Physical and Engineering Sciences in Medicine, 40 (2), pp. 289-295.
Reviewer 3 Report
" Second generation of tissue-engineered ligament substitutes for torn ACL replacement: adaptations for clinical applications"
It is interesting to investigate if acellular bACLs can be used to replace torn ACLs. However, there are some corrections that are essential to meet the standard for publication. Please refer to the following comments.
There are some problems with how to write your research paper.
Especially in the "Materials and Methods" section.
1) Please add the Institutional Review Board approval number and approval date and time for animal testing.
2) Please add the type and method of test drug used in the section "Histological analysis of bACLs before and after implantation, ex-vivo".
3) Please add the specific method in the "Mechanical analyzes" section. If you are concerned that your manuscript will be too long, we recommend using the auxiliary fields.
4) In the "Surgical testing" section, please add what you specifically evaluated and how.
5) Please add the limitation of your research in the discussion. How did you determine the number of validations for this study? Is this number of verifications appropriate?
6) Many of the cited references are old. There are many references that are more than 10 years old, so please update to the latest paper as a whole.
Round 2
Reviewer 1 Report
Based on the comments from the authors, most of the questions have been solved. I think it is possible to get it published after careful language editing.
Reviewer 3 Report
Thank you for giving me this opportunity to re-review your revised manuscript.
I am happy that all of the suggested corrections have been made.
Thank you for spending so much time for revised manuscript.